

# Assessment of health promotion behavior and associated factors among the northern Saudi adolescent population: a cross-sectional study

Thamer Alshami M. Alruwaili[1],  Saad Abdullah K. Alshehri[2],
Ashokkumar Thirunavukkarasu[3],  Mohamed Shawky Elfarargy[1,4],
Khalid Tariq Alanazi[5],  Khalid Muharib R. Alruwaili[5],
Yousef Salman Abdullah Alanezi[5] and  Abdulelah Abdulhadi Alruwaili[5]

[1] Department of Pediatrics, College of Medicine, Jouf University, Sakaka, Aljouf, Saudi Arabia
[2] Commitment Department, Ministry of Health, Directorate of Health Affairs, Riyadh, Saudi Arabia
[3] Department of Community and Family Medicine, College of Medicine, Jouf University, Sakaka, Aljouf, Saudi Arabia
[4] Department of Pediatrics, Faculty of Medicine, Tanta University, Gharbia Governorate, Egypt
[5] College of Medicine, Jouf University, Sakaka, Aljouf, Saudi Arabia

## ABSTRACT

**Background and Aim**.  Health promotions among the adolescent population have a significant role in achieving the 2030 sustainable development goals of the World Health Organization. The COVID-19 pandemic has led to several devastating impacts on the health, economic, social, and healthcare systems, and adolescents' health promotions are no exception. We assessed health promotion behaviors and associated factors among the adolescent population of northern Saudi Arabia (KSA).

**Methods**. We used the Arabic version of the adolescent health promotion scale (AHPS-40) among the 400-adolescent population. The AHPS-40 assessed six domains of adolescent health behavior: nutrition, social support, health responsibility, life appreciation, exercise, and stress management. We applied the Chi-square test to identify the associated factors of adolescent health promotion activities and the logistic regression test to find the predictors for overall health promotion categories.

**Results**. Of the studied participants, the mean $\pm$ SD of the total AHPS-40 was $103.31 \pm 18.78$. The nutrition domain of the AHPS-40 was significantly associated with the age group ($p = 0.002$), and the social support domain was significantly related to fathers' ($p = 0.022$) and mothers' education ($p = 0.006$). The exercise domain of AHPS-40 was significantly associated with age group ($p = 0.018$) and school level ($p = 0.026$). Gender was significantly associated with most of the six domains. Furthermore, more than half (52.7%) of them had a low health promotion behavior, which was significantly associated with gender (adjusted odds ratio $= 1.59$, 95% CI of AOR $= 1.04 - 2.45$, $p = 0.032$).

**Conclusion**. Our study results suggest improving health promotion behaviors by instituting awareness-raising and health promotion intervention programs for adolescent groups. Furthermore, we recommend a focused, exploratory, mixed-method survey among the adolescents of other regions of KSA to identify the region-specific adolescent's health promotion behaviors.

Corresponding author
Thamer Alshami M. Alruwaili,
tsruwaili@ju.edu.sa

# INTRODUCTION

The World Health Organization (WHO) defines adolescence as the transition phase from childhood to adulthood, spanning from 10 to 19 years of age. Health promotions among the adolescent population have a significant role in achieving the 2030 sustainable development goals set by the WHO (*WHO, 2022a*). The adolescence period is challenging for growth and development as the requirements for psychological, physical, and emotional needs are unique to this age group (*Alderman & Breuner, 2019*; *Vazquez-Ortiz et al., 2020*). Notably, the transformation phase from childhood to adulthood poses a more critical public health challenge on adolescents' life than ever before due to the influence of social media, peer forces, and COVID-19 pandemic (*Buda et al., 2021*; *Oliveira et al., 2020*; *Paakkari et al., 2021*).

Health promotion is often considered one of the effective tools in global public health activities (*Kumar & Preetha, 2012*; *Pati et al., 2017*). In the context of adolescent care, health promotion is defined as "A continuous process that facilitates adolescents to choose healthy behaviors and to improve control over their health" (*WHO, 2022c*). The health promotion approaches and strategies for adolescents include health education, life skills development, policy changes, a supportive environment, community involvement, and provision of accessible healthcare services (*Philip, Kannan & Parambil, 2018*; *Salam et al., 2016*; *WHO, 2022c*). Globally, adolescents contribute to 16% of the total population. However, their needs are often ignored due to the apparently healthy people and comparatively low incidence of mortality in this group (*WHO, 2022a*).

Several authors stated that common health promotion areas to be focused on in adolescents are physical activities, nutrition, mental health, life skills, social involvement, tobacco and alcohol use, drug abuse, sexual health, and risk-taking behaviors (*CDC, 2019*; *Eschenbeck et al., 2019*; *Salam et al., 2016*; *Viner & Macfarlane, 2005*). The majority of the evidence suggests that healthy behaviors related to physical and dietary activities established in childhood and adolescence might track into adulthood (*Barrington-Trimis et al., 2020*; *Viner & Macfarlane, 2005*; *Yang et al., 2020*). For example, adolescent obesity is an important predictor of the development of obesity in adults. A recently concluded follow-up study from the USA reported that adolescents who experienced obesity at the age of 15–17 had a higher positive predictive value for developing obesity at 50 (*Rundle et al., 2020*). It is a well-established fact that obesity is a major and modifiable risk factor for type 2 diabetes mellitus, hypertension, and ischemic heart disease.

The COVID-19 pandemic has led to several devastating impacts on the health, economic, social, and healthcare systems, and adolescents are no exception. Even though adolescents might have a lower risk of developing COVID-19-related complications, the pandemic has disturbed their routine activities due to lockdowns, school closures, online lectures, and separation from their friends and peers. As per the Johns Hopkins Center for Adolescent Health, the COVID-19 prevention strategies, namely face masks, social distancing, and

interruption in regular school activities might have various mental and emotional effects on adolescents than adults (*Volkin, 2020*). A report by the United Nations Children's Fund (UNICEF) revealed that nearly half of the young people had less motivation to do their regular activities, which they generally enjoyed, and 43% of adolescent girls were pessimistic about the future (*UNICEF, 2021*).

A 2021 study by *Stavridou et al. (2021)* reported that during the COVID-19 period, children, adolescents, and young adults gained weight significantly. Furthermore, adolescents also developed unhealthy dietary behaviors, including excessive consumption of potatoes, meats, and carbonated beverages. Another survey conducted in Italy by *Pietrobelli et al. (2020)* reported that sleep, electronic screen usage, and potato intake had increased significantly, while physical activities have decreased. These unhealthy behaviors were substantially higher among adolescents of lower age and male gender (*Donati et al., 2021*) and were substantially higher among adolescents of lower age and female genders. On assessing health promotion behaviors of adolescents before the COVID-19 pandemic, *Musavian et al. (2014)* found a significant association between adolescent health promotion scores with age, gender, and father's and mother's educational level. According to a study by *Chen, James & Wang (2007)* healthcare professionals should pay close attention to adolescents' dietary habits, exercise, social support, stress management abilities, appreciation of life, and health responsibility to support their lifestyle. A health-promoting behavior and most unhealthy lifestyle habits are established during adolescence, as stated by *Wang et al. (2009)*. Therefore, assessing health-promoting behavior among adolescents has become a research focus worldwide, including in the KSA. So that necessary health programs for adolescents can be implemented by the policymakers. The prevalence of modifiable risk factors and poor health promotion is very high among Saudi adolescents. For example, *Al-Hazzaa & Albawardi (2021)* reported that about 40% of Saudi adolescents were either overweight or obese. Continuous evaluation of health promotion behaviors, including nutrition, physical activities, social responsibility, and stress management, is critical to plan and implementing the required care services for adolescents at different levels. However, from the current study authors' extensive literature review, there is limited data on the northern region of the Kingdom of Saudi Arabia (KSA). Taking into the account of having region-specific data, the present study is planned to assess health promotion behaviors and associated factors among the adolescent population of northern KSA. The study utilized the Arabic version of the adolescent health promotion scale (AHPS-40) to assess six domains of adolescent health behavior, namely nutrition, social support, health responsibility, life appreciation, exercise, and stress management.

## MATERIALS & METHODS

### Study description

The present study was conducted from October 2022 to January 2023 in the Aljouf province of the KSA. Aljouf province is located in the northern part of the KSA, with a total population is approximately half a million. This province has four administrative regions: Sakaka, Duma al Jindal, Tabarjil, and Qurrayat. The present study included an

adolescent population aged between 10 and 19 from all four regions. The demographic data from the General Authority for Statistics, KSA, indicated that the total adolescent in this region is about 80 thousand, most of which belong to Saudi nationals. The age group was set according to the WHO classifications. This study utilized a cross-sectional survey research design with a one-shot data collection approach. Furthermore, data were collected from a sample of 400 adolescents in northern Saudi Arabia using the Arabic version of the AHPS-40. The AHPS-40 assessed six domains of adolescent health behavior, namely nutrition, social support, health responsibility, life appreciation, exercise, and stress management.

## Sample size estimation and sampling procedure

We calculated the required sample for this study based on Cochran's sample size estimation equation ($n = z2\ pq\ /\ e2$). In this formula, n is the number of participants required, $z = 1.96$ in 95% confidence interval, p = expected proportion, $q = 1 - p$, and e is 5% of the margin of error. Since previous research on adolescents revealed different prevalence on this subject, we have taken 50% as the expected proportion to obtain the maximum sample size. Applying the values mentioned above in Cochran's equation, the estimated sample size for the present study was 384 and rounded to 400. A consecutive sampling method was applied to select study participants. In this method, we invited adolescents from public places like shopping malls, parks, and supermarkets.

## Inclusion and exclusion criteria

The present study included all the Saudi adolescents as per the WHO criteria, from Aljouf province, and those accompanied by parent (s) or legal guardian (if less than 18 years old) were invited to participate in this study. Those outside Aljouf province and with the existing physically and mentally disabled adolescents were excluded from the survey.

## Data collection procedure

The current study was given ethical clearance from the Jouf University Ethical Committee, Saudi Arabia (Approval no: 6-02-44, Dated: 13th October 2022). The research team started the study data collection process after briefing the study purposes to the parent (s)/legal guardians and adolescents. After obtaining written informed consent from the parent or legal guardian, we requested the participant to fill out a pretested data collection proforma (no identity details were collected) consisting of two sections. In the first section, we inquired about the participants' background characteristics such as age, gender, academic performance in the previous term, and father's and mother's education. In the second part, we used the AHPS-40, a valid and reliable tool developed by *Chen et al. (2003)*. The original instrument had an overall high internal consistency (Cronbach's alpha = 0.93), and the Cronbach's alpha values of the six domains ranged from 0.75 to 0.88. It was translated and used in several other countries to assess the health promotion behaviour of adolescents (*Guedes & Zuppa, 2020*; *Ortabag et al., 2011*). The AHPS-40 is a self-report tool consists of 40 questions that are further divided into six subsections, namely nutrition (6-items), social support (7-items), health responsibility (8-items), life appreciation (8-items), exercise (5-items), and stress management (6-items). We have translated the questionnaire into

Arabic using the standard translation protocols (*Robichaud-Ekstrand, Haccoun & Millette, 1994*; *Tsang, Royse & Terkawi, 2017*). We pretested the questionnaire on the 30-adolescent population during the pilot study. Before the pilot study, the research team received clearance from the Jouf University review board. All the adolescents involved in the pretest (pilot survey) indicated that the data collection tool was simple and easy to interpret. Furthermore, the Cronbach analysis revealed that the scores of all six subsections were within the acceptable range (more than 0.70). Furthermore, the result of the Kaisor-Meyer-Olkin was about 0.8 and Bartlett's sphericity test was <0.001 (chi-square value = 3125.33, $df = 780$, and $p = 0.0001$) (essential criteria to do Exploratory Factor Analysis (EFA)). We presented the values of EFA in the table attached in the supplementary section (Exploratory Factor Analysis, Supplementary File). The participants responded on a 5-point Likert's response scale with any of the following answers, "always", "usually", "sometimes", "rarely", or "never", and these responses were scored from 5 to 1. Hence, the possible total score for all the responses ranged from 40 to 200.

## Data analysis

The research team used Statistical Package for Social Science (SPSS, V.21) for data curation and analysis. The descriptive information of the participated adolescents is depicted as number, proportion, and mean ± standard deviation (SD). We calculated the mean (±SD) of all six domains of AHPS-40 and the total score of AHPS-40. The scores below the mean and above the mean were considered low and high for adolescents' health promotion. Furthermore, we applied the Chi-square test to identify the associated factors of adolescent health promotion activities. Finally, we executed binomial logistic regression statistical analysis to find the predictors for overall health promotion scores. We have set a two-tailed $p$-value ($<0.05$) as a statistically significant association.

## RESULTS

The present study participants' background characteristics are presented in Table 1. Of the four hundred (400) participated adolescents, 38.0% belonged to the late adolescent's category, 52.2% were girls, and the majority (44.5%) were studying high school and above.

The median and interquartile range (IQR) of the six domains (nutrition (15 and 5), social support (17 and 6), health responsibility (22 and 6), life appreciation (16.5 and 8), exercise (15 and 7), stress management (16 and 6)) and total AHPS-40 (102 and 24) are depicted in Table 2.

The nutrition domain of the AHPS-40 was significantly associated with age group ($p = 0.002$), gender ($p = 0.007$), and gender ($p = 0.001$). The social support domain was significantly related to fathers' ($p = 0.022$) and mothers' education ($p = 0.006$) (Table 3).

Table 4 represents the participants' background characteristics with health responsibility and life appreciation domains. The age group of the responded participants is significantly associated with health responsibility ($p = 0.024$) and life appreciation ($p = 0.004$) domains.

The exercise domain of AHPS-40 was significantly associated with age group ($p = 0.018$), gender ($p = 0.001$), school level ($p = 0.026$), and mother's education ($p = 0.041$). The stress

**Table 1** **Background characteristics of the adolescents ($n = 400$).** Each background characteristics are presented as frequency ($n$) and proportion (%).

| Variables | Frequency | Proportion |
|---|---|---|
| Age group | | |
| Early adolescent | 119 | 29.7 |
| Middle adolescent | 129 | 32.3 |
| Late adolescent | 152 | 38.0 |
| Gender | | |
| Boys | 191 | 47.8 |
| Girls | 209 | 52.2 |
| Currently studying class (school) | | |
| High school and above | 178 | 44.5 |
| Middle/Intermediate | 155 | 38.8 |
| Primary | 67 | 16.7 |
| Grade in the previous semester | | |
| Excellent | 288 | 72.0 |
| Very good | 62 | 15.5 |
| Good | 50 | 12.5 |
| Fathers' education | | |
| Above high school | 205 | 51.3 |
| Up to high school | 195 | 48.7 |
| Mothers' education | | |
| Above high school | 187 | 46.8 |
| Up to high school | 213 | 53.2 |

**Table 2** **The median and Interquartile range (IQR) scores of all six domains and total AHPS-40.** Each subscales of AHPS-40 is presented as median and IQR.

| Item | Median | IQR |
|---|---|---|
| Nutrition | 15 | 5 |
| Social support | 17 | 6 |
| Health responsibility | 22 | 6 |
| Life appreciation | 16.5 | 8 |
| Exercise | 15 | 7 |
| Stress management | 16 | 6 |
| Total AHPS-40 score | 102 | 24 |

management domain was significantly associated with the participants' gender ($p = 0.029$) (Table 5).

The mean score of the total AHPS-40 of the present study was 103.31. Of the 400 participants, less than half (47.3%) of them had a high health promotion behavior (above the mean total score of AHPS-40). Otherwise, they were categorized as being in the lower group. Through the binomial logistic regression analysis (after adjusting with other covariables), we found a significant association with the gender of the participants (adjusted

**Table 3 Association between adolescents' background characteristics with the nutrition and social support domains (n = 400).** The associated factors for nutrition and social support domains are analyzed and depicted with significance (p < 0.05).

| Variable | Total | Nutrition | | | Social support | | |
|---|---|---|---|---|---|---|---|
| | | Low n (%) | High n (%) | p-value | Low n (%) | High n (%) | p-value |
| Age group | | | | | | | |
| Early adolescent | 119 | 75 (63.0) | 44 (37.0) | 0.002* | 55 (46.2) | 64 (53.8) | 0.108 |
| Middle adolescent | 129 | 70 (54.3) | 59 (45.7) | | 76 (58.9) | 53 | |
| Late adolescent | 152 | 63 (41.4) | 89 (58.6) | | 75 (49.3) | 77 (50.7) | |
| Gender | | | | | | | |
| Boys | 191 | 113 (59.2) | 78 (40.8) | 0.007* | 100(52.4) | 91(47.6) | 0.743 |
| Girls | 209 | 95 (45.5) | 114 (54.5) | | 106 (50.7) | 103 (49.3) | |
| Currently studying class (School) | | | | | | | |
| High school and above | 178 | 110 (61.8) | 68 (38.2) | 0.001* | 94 (52.8) | 84 (47.2) | 0.212 |
| Middle/Intermediate | 155 | 73 (47.1) | 82 (52.9) | | 84 (54.2) | 71 (45.8) | |
| Primary | 67 | 25 (37.3) | 42 (62.7) | | 28 (41.8) | 39 (58.2) | |
| Grade in the previous semester | | | | | | | |
| Excellent | 288 | 156 (54.2) | 132 (45.8) | 0.302 | 156 (54.2) | 132 (45.8) | 0.224 |
| Very good | 62 | 27 (43.5) | 35 (56.5) | | 27 (43.5) | 35 (56.5) | |
| Good | 50 | 25 (50.0) | 25 (50.0) | | 23 (46.0) | 27 (54.0) | |
| Fathers' education | | | | | | | |
| Above high school | 205 | 109 (53.2) | 96 (46.8) | 0.689 | 117 (57.1) | 88 (42.9) | 0.022* |
| Up to high school | 195 | 99 (50.8) | 96 (49.2) | | 89 (45.6) | 106 (54.4) | |
| Mothers' education | | | | | | | |
| Above high school | 187 | 106 (56.7) | 81 (43.3) | 0.079 | 110 (58.8) | 77 (41.2) | 0.006* |
| Up to high school | 213 | 102 (47.9) | 111 (52.1) | | 96 (45.1) | 117 (54.9) | |

**Notes.**
*Significant value obtained through Chi-square test.

odds ratio (AOR) = 1.59, 95% CI of AOR = 1.04–2.45, p = 0.032). No other background characteristics had a significant association (Table 6).

## DISCUSSION

The global goals for sustainable development made an urgent call to action for "adolescent health promotion and well-being" as a priority to achieve the sustainable development goal (SDG) target 3.8. This reinstates the importance of adolescent health promotion activities globally to achieve the United Nations SDG targets (WHO, 2022b). The present study assessed health promotion behavior and associated factors among the northern Saudi adolescent population using the AHPS-40.

The present study found that the mean score of the total AHPS-40 was around 103, and only 47.3% of the participants had high health promotion activities. A study conducted by Chen et al. (2003) using AHPS-40 reported that their respondents had a higher mean score (129) than our study, and the range of their participants' scores were from 51 to 176. Similarly, another study conducted in Turkey by Ozturk & Ayaz-Alkaya (2020) reported a higher mean AHPS score than the present study. The most likely reason for the differences in study settings. Furthermore, our study was performed during the COVID-19 era, and

**Table 4  Association between adolescents' background characteristics with the health responsibility and life appreciation domains ($n = 400$).** The associated factors for health responsibility and life appreciation are analyzed and depicted with significance ($p < 0.05$).

| Variable | Total | Health responsibility | | | Life appreciation | | |
|---|---|---|---|---|---|---|---|
| | | Low $n$ (%) | High $n$ (%) | $p$-value | Low $n$ (%) | High $n$ (%) | $p$-value |
| Age group | | | | | | | |
| Early adolescent | 119 | 76 (63.9) | 43 (36.1) | 0.024[*] | 52 (43.7) | 67 (56.3) | 0.004[*] |
| Middle adolescent | 129 | 68 (52.7) | 61 (47.3) | | 77 (59.7) | 52 (40.3) | |
| Late adolescent | 152 | 72 (47.4) | 80 (52.6) | | 96 (63.2) | 56 (36.8) | |
| Gender | | | | | | | |
| Boys | 191 | 106 (55.5) | 85 (44.5) | 0.566 | 96 (50.3) | 95 (49.7) | 0.021[*] |
| Girls | 209 | 110 (52.6) | 99 (47.4) | | 129 (61.7) | 80 (38.3) | |
| Currently studying class (School) | | | | | | | |
| High school and above | 178 | 106 (59.6) | 72 (40.4) | 0.129 | 90 (50.6) | 88 (49.4) | 0.119 |
| Middle/Intermediate | 155 | 78 (50.3) | 77 (49.7) | | 95 (61.3) | 60 (38.7) | |
| Primary | 67 | 32 (47.8) | 35 (52.2) | | 40 (59.7) | 27 (40.3) | |
| Grade in the previous semester | | | | | | | |
| Excellent | 288 | 159 (55.2) | 129 (44.8) | 0.620 | 168 (58.3) | 120 (41.7) | 0.338 |
| Very good | 62 | 30 (48.4) | 32 (51.6) | | 30 (48.4) | 32 (51.6) | |
| Good | 50 | 27 (54.0) | 23 (46.0) | | 27 (54.0) | 23 (46.0) | |
| Fathers' education | | | | | | | |
| Above high school | 205 | 115 (56.1) | 90 (43.9) | 0.388 | 115 (56.1) | 90 (43.9) | 0.950 |
| Up to high school | 195 | 101 (51.8) | 94 (48.2) | | 110 (56.4) | 85 (43.6) | |
| Mothers' education | | | | | | | |
| Above high school | 187 | 112 (59.9) | 75 (40.1) | 0.027 | 113 (60.4) | 74 (39.6) | 0.115 |
| Up to high school | 213 | 102 (48.8) | 109 (51.2) | | 112 (52.6) | 101 (47.4) | |

**Notes.**
*Significant value obtained through Chi-square test.

the latter studies were done before the COVID-19 pandemic. This finding also affirms the impact of the COVID-19 pandemic on adolescent health and well-being (*Ashwin, Cherukuri & Rammohan, 2022*; *Branje & Morris, 2021*).

The adolescent period is a nutrition-sensitive phase of growth and development. Proper nutrition during adolescence plays a significant role in puberty, muscle, and fat mass, and preventing several non-communicable diseases (*Das et al., 2017*; *Norris et al., 2022*). Nonetheless, during the COVID-19 pandemic and lockdown period, the nutritional habits of children and adolescents changed dramatically (*Pujia et al., 2021*; *Teixeira et al., 2021*). The present study found that nearly half of the adolescents' health promotion behavior was low and significantly associated with age groups, school type, and gender. Interestingly, another study done in the middle east region also reported a similar association for the nutrition domain with gender ($p < 0.001$) and school grade ($p = 0.042$) (*Musavian et al., 2014*). In contrast, *Ortabag et al. (2011)* did not find an association between gender and the nutrition domain. The possible variations between our study findings and *Ortabag et al. (2011)* could be study settings, inclusion, and exclusion criteria.

Empowering adolescents and their responsibility towards healthy behaviors are the key components of adolescent periods (*Moilanen et al., 2018*). The present study found

**Table 5 Association between adolescents' background characteristics with the exercise and social stress management domains ($n = 400$).** The associated factors for the exercise and social stress management are analyzed and depicted with significance ($p < 0.05$).

| Variable | Total | Exercise | | | Stress management | | |
|---|---|---|---|---|---|---|---|
| | | Low $n$ (%) | High $n$ (%) | $p$-value | Low $n$ (%) | High $n$ (%) | $p$-value |
| Age group | | | | | | | |
| Early adolescent | 119 | 78 (65.5) | 41 (34.5) | 0.018[*] | 65 (54.6) | 54 (45.4) | 0.350 |
| Middle adolescent | 129 | 68 (52.7) | 61 (47.3) | | 59 (45.7) | 70 (54.3) | |
| Late adolescent | 152 | 74 (48.7) | 78 (51.3) | | 73 (48.0) | 79 (52.0) | |
| Gender | | | | | | | |
| Boys | 191 | 134 (70.2) | 57 (29.8) | 0.001[*] | 105 (55.0) | 86 (45.0) | 0.029[*] |
| Girls | 209 | 86 (41.1) | 123 (58.9) | | 92 (44.0) | 117 (56.0) | |
| Currently studying class (School) | | | | | | | |
| High school and above | 178 | 111 (62.4) | 67 (37.6) | 0.026[*] | 98 (55.1) | 80 (44.9) | 0.109 |
| Middle/Intermediate | 155 | 78 (50.3) | 77 (49.7) | | 68 (43.9) | 87 (56.1) | |
| Primary | 67 | 31 (46.3) | 36 (53.7) | | 31 (46.3) | 36 (53.7) | |
| Grade in the previous semester | | | | | | | |
| Excellent | 288 | 150 (52.1) | 138 (47.9) | 0.098 | 145 (50.3) | 143 (49.7) | 0.778 |
| Very good | 62 | 36 (58.1) | 26 (41.9) | | 29 (46.8) | 33 (53.2) | |
| Good | 50 | 34 (68.0) | 16 (31.0) | | 23 (46.0) | 27 (54.0) | |
| Fathers' education | | | | | | | |
| Above high school | 205 | 113 (55.1) | 92 (44.9) | 0.960 | 110 (48.8) | 105 (51.2) | 0.847 |
| Up to high school | 195 | 107 (54.9) | 88 (45.1) | | 97 (49.7) | 98 (50.3) | |
| Mothers' education | | | | | | | |
| Above high school | 187 | 113 (60.4) | 74 (39.6) | 0.041[*] | 98 (52.4) | 89 (47.6) | 0.237 |
| Up to high school | 213 | 107 (50.2) | 106 (49.8) | | 99 (46.5) | 114 (53.5) | |

**Notes.**

*Significant value obtained through Chi-square test.

a significant association between age group and health responsibility behavior of the responded adolescents. Regarding the life appreciation domain, we found a significant association between the age group and gender. A correlation study conducted by *Ayres & Pontes (2018)* found a positive correlation between the health-promoting behavior of adolescents and health-responsibility behavior. Iranian research revealed a significant association between health responsibility and gender and school level (*Musavian et al., 2014*). Similarly, a study conducted in Turkey revealed a positive association of health responsibility with gender and school level (*Ortabag et al., 2011*).

Our study findings explored that the exercise domain is significantly associated with age group, gender, school level, and mothers' education. Interestingly, gender was one of the critical predictors of physical activity among adolescents before and during COVID-19 era. A recent survey conducted among adolescents in the Jeddah region of KSA stated that males were more likely to be physically active than females (*Almutairi, Burns & Portsmouth, 2022*). Similar to the present study, *Musavian et al. (2014)* also found a significant association with gender. Since physical inactivity is directly linked with the mental health status of adolescents, the present study found a significant association between genders in stress management. The present study found that gender is one of the important predictors

**Table 6  Predictors for the low health promotion among the northern Saudi adolescents ($n = 400$).** Each variables are evaluated with the outcome variable (total AHPS-40) using the logistic regression.

| Variable | Total | AHPS-40 | | | |
|---|---|---|---|---|---|
| | | Low ($n = 211$) | High ($n = 189$) | Adjusted odds ratio[*] (95% CI) | *p*-value |
| Age group | | | | | |
| Early adolescent | 119 | 68 | 51 | Ref | |
| Middle adolescent | 129 | 68 | 61 | 0.89 (0.48–1.61) | 0.691 |
| Late adolescent | 152 | 75 | 77 | 0.84 (0.43–1.66) | 0.617 |
| Gender | | | | | |
| Boys | 191 | 113 | 78 | Ref | 0.032[**] |
| Girls | 209 | 98 | 111 | 1.59 (1.04–2.45) | |
| Currently studying class (School) | | | | | |
| High school and above | 178 | 104 | 74 | Ref | |
| Middle/Intermediate | 155 | 75 | 80 | 1.46 (0.82–2.26) | 0.201 |
| Primary | 67 | 32 | 35 | 1.49 (0.72–3.11) | 0.278 |
| Grade in the previous semester | | | | | |
| Excellent | 288 | 157 | 131 | Ref | |
| Very good | 62 | 28 | 34 | 1.47 (0.83–2.62) | 0.187 |
| Good | 50 | 26 | 24 | 1.16 (0.61–2.22) | 0.656 |
| Fathers' education | | | | | |
| Above high school | 205 | 112 | 93 | Ref | 0.916 |
| Up to high school | 195 | 99 | 96 | 0.98 (0.64–1.49) | |
| Mothers' education | | | | | |
| Above high school | 187 | 109 | 78 | Ref | 0.125 |
| Up to high school | 213 | 102 | 111 | 1.39 (0.91–2.12) | |

**Notes.**

[*]Adjusted variables: Age group, Gender, School, Grade in previous semester, fathers' education, mother's education.

[**]Significant *p*-value.

of health promotion behavior (AOR = 1.59, 95% CI of AOR = 1.04–2.45). A study conducted among Brazilian adolescents reported a mothers' education and male gender were significantly associated with health-related behaviors (*Azeredo et al., 2016*). The possible difference between our study and the Brazilian study could be the differences in sociocultural variations.

Even though the present study is the first of its kind in the northern KSA using a standard and validated instrument with an adequate sample size, some constraints cannot be excluded while reading the present survey. Firstly, we utilized the consecutive method to invite adolescents to participate in the present study. Secondly, we executed the present population-based study in the northern regions of the KSA. Hence, the present study findings may not be representative findings of total KSA due to immense cultural differences across the country. Finally, our cross-sectional study on adolescents evaluated the association between variables, not the causation.

## CONCLUSIONS

The present survey findings revealed that northern Saudi adolescents' health promotion behaviors are low in all six domains of AHPS-40. We also found several background characteristics associated with the health promotion behaviors of adolescents. The age group was the significantly associated factor for most health promotion domains except social support and stress management. Our study results suggest improving the health promotion behaviors of the adolescent population. We also recommend instituting awareness-raising and health promotion intervention programs for adolescent groups, which are found to have significantly low health promotion behaviors. Also, these programs are to be targeted toward specific domains. Furthermore, we recommend a focused, exploratory, mixed-method survey among the adolescents of other regions of KSA to identify the region-specific adolescent's health promotion behaviors and plan the required policy to improve their health promotion behavior.

## ACKNOWLEDGEMENTS

The research team wishes to thank all participants and parents for their participation in the survey.

### Funding

This work was funded by the Deanship of Scientific Research at Jouf University under grant No (DSR2022-NF-18). The funders had no role in study design, data collection and analysis, decision to publish, or preparation of the manuscript.

### Grant Disclosures

The following grant information was disclosed by the authors:
Deanship of Scientific Research at Jouf University: DSR2022-NF-18.

### Competing Interests

The authors declare there are no competing interests.

### Author Contributions

- Thamer Alshami M. Alruwaili conceived and designed the experiments, performed the experiments, prepared figures and/or tables, authored or reviewed drafts of the article, getting necessary other approvals, and approved the final draft.
- Saad Abdullah K. Alshehri conceived and designed the experiments, performed the experiments, analyzed the data, prepared figures and/or tables, and approved the final draft.
- Ashokkumar Thirunavukkarasu conceived and designed the experiments, analyzed the data, prepared figures and/or tables, back translation, and approved the final draft.
- Mohamed Shawky Elfarargy conceived and designed the experiments, prepared figures and/or tables, authored or reviewed drafts of the article, and approved the final draft.

- Khalid Tariq Alanazi performed the experiments, authored or reviewed drafts of the article, and approved the final draft.
- Khalid Muharib R. Alruwaili performed the experiments, prepared figures and/or tables, authored or reviewed drafts of the article, and approved the final draft.
- Yousef Salman Abdullah Alanezi performed the experiments, authored or reviewed drafts of the article, and approved the final draft.
- Abduelah Abdulhadi Alruwaily conceived and designed the experiments, prepared figures and/or tables, authored or reviewed drafts of the article, and approved the final draft.

## Human Ethics

The following information was supplied relating to ethical approvals (*i.e.*, approving body and any reference numbers):

The Local Committee of Bioethics, Jouf University, Saudi Arabia approved the study (Approval no: 6-02-44, Dated: 13th October 2022).

## Data Availability

The raw data used for analysis is available in the Supplemental File.

## Supplemental Information

Supplemental information for this article can be found online at http://dx.doi.org/10.7717/peerj.15567#supplemental-information.

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
