# Peer review of "Assessment of health promotion behavior and associated factors among the northern Saudi adolescent population: a cross-sectional study"

_PeerJ, doi:10.7717/peerj.15567_

## Round 0.1 · original submission · Major Revisions

Thank you for your submission. The reviewers have identified a number of concerns that must be addressed. In particular in validating the instrument and the statistics used.

Reviewer 1 ·

Basic reporting

Based on my review on the research design of this current study, there were some weakness that could be improved for the next review:
a) The writing on the method was too brief, which the authors could describe/explain more details on the demographics aspects of the sample.
b) The authors could explain in details about the research design of this study, whether it was a one-shot data collection for survey research design.
c) The authors failed to state what are the research questions of this study, and the problem statements (gap of study) were not too clear in this article.

Experimental design

In term of research design, I found out some major flaws of this study:
a) The authors did not conduct a vigorous validation on the instrument, especially on the "construct validity" procedure. At this level of the study, which consisted a large number of participants were involved, at least the authors/researchers could provide an EFA (Exploratory Factorial Analysis), or perhaps CFA (Confirmatory Factorial Analysis) output. Without providing such figures or conducting such procedures (e.g., EFA or CFA), it will lead to a weak internal validity of the instrument.

Validity of the findings

I'm wondering WHY the authors utilized the Chi-square or OR analysis instead, Spearmen's correlation is suffice to be used as analysis in this study. I have no objection about the logistic regression modeling that have been used by the authors, but at least please report the appropriate effect size and how much explained variance from this model predicted (predictors) on the outcome variables.

Additional comments

No comment

Reviewer 2 ·

Basic reporting

abstract:
Clear and unambiguous: The abstract provides a clear overview of the study's purpose, methods, and main findings. The language used is mostly clear and unambiguous, although there are some areas where the language could be improved (e.g., "the present study assessed" could be changed to "we assessed").
Professional English used throughout: The language used is generally professional.
Self-contained with relevant results to hypotheses: The abstract provides relevant results to the study's hypotheses, including associations between certain domains of adolescent health behavior and factors such as age group, gender, and parental education. The conclusion also suggests areas for future research.
Overall, the abstract provides a clear and concise summary of the study's purpose, methods, and main findings.

Experimental design

the abstract appears to be within the scope of the Health Sciences and Medical Sciences, which are areas that PeerJ considers for publication.

the materials and methods appear to be appropriate for the study. The study provides a clear description of the study population, sample size estimation, and sampling procedure. The inclusion and exclusion criteria are also well-defined. The data collection procedure is ethical, and the questionnaire used has been translated using standard protocols and tested for reliability and validity. The statistical methods used for data analysis are appropriate and relevant to the research questions being asked.

Validity of the findings

the conclusion is well-stated and linked to the original research question. The research question is related to exploring the health promotion behaviors of adolescents in northern Saudi Arabia, and the conclusion provides a clear summary of the study's findings related to this question. The conclusion also offers recommendations based on the results to improve adolescent health promotion behaviors and suggests further research to explore the health promotion behaviors of adolescents in other regions of Saudi Arabia. The conclusion is limited to supporting the results and does not make any claims beyond what was found in the study.

Additional comments

1- table 1 the proportions can mention in () and combined by table 2. table 2 SD should be followed by +/- sign. Table 3-5 is the distribution and not relationship.
2- if any variable is not normally distributed use median (IQR) not mean +/- SD
3- how you selected criteria for logistic regression in table 6?

---

## Round 0.2 · Minor Revisions

Thank you for your revised submission. One of the reviewers still has concerns about how how you report EFA. Please strengthen this aspect of the work accordingly.

Reviewer 1 ·

Basic reporting

In general, the basic reporting is satisfied for the purpose to publish in this article.

Experimental design

Not applicable.

Validity of the findings

Despite, the authors have stated it in the rebuttal statements about the EFA values for the purpose of construct validity. I'm suggesting the authors to report it in details (the values of EFA) specifically in the validation section (method) of this article.

Additional comments

Overall, I do satisfied with all the amendments that have been done by the authors. Congratulation!

Reviewer 2 ·

Basic reporting

thank you very much
I have no further comments

Experimental design

I have no further comments

Validity of the findings

I have no further comments

Additional comments

I have no further comments

---

## Round 0.3 · accepted · Accept

Thank you for your revised submission.